# A New Family of Homoleptic Copper Complexes of Curcuminoids: Synthesis, Characterization and Biological Properties

**DOI:** 10.3390/molecules24050910

**Published:** 2019-03-05

**Authors:** William Meza-Morales, Juan C. Machado-Rodriguez, Yair Alvarez-Ricardo, Marco A. Obregón-Mendoza, Antonio Nieto-Camacho, Rubén. A. Toscano, Manuel Soriano-García, Julia Cassani, Raúl G. Enríquez

**Affiliations:** 1Instituto de Química, Universidad Nacional Autónoma de México, Circuito Exterior, Ciudad Universitaria, Mexico City C.P. 07340, Mexico; willy_meza_morales@hotmail.com (W.M.-M.); jcmrn.n@gmail.com (J.C.M.-R.); yfar30@hotmail.com (Y.A.-R.); obregonmendoza@yahoo.com.mx (M.A.O.-M.); camanico2015@yahoo.com (A.N.-C.); toscano@unam.mx (R.A.T.); soriano@unam.mx (M.S.-G.); 2Departamento de Sistemas Biológicos, Universidad Autónoma Metropolitana, Unidad Xochimilco, Mexico City C.P. 04960, Mexico

**Keywords:** crystal structure, homoleptic copper complexes, curcuminoids, antioxidant activity

## Abstract

We report herein the synthesis and crystal structures of five new homoleptic copper complexes of curcuminoids. The scarcity of reports of homoleptic complex structures of curcuminoids is attributed to the lack of crystallinity of such derivatives, and therefore, their characterization by single crystal X-ray diffraction is rare. The ligand design suppressing the phenolic interaction by esterification or etherification has afforded a significant increase in the number of known crystal structures of homoleptic metal complexes of curcuminoids revealing more favorable crystallinity. The crystal structures of the present new copper complexes show four-fold coordination with a square planar geometry. Two polymorphs were found for DiBncOC-Cu when crystallized from DMF. The characterization of these new complexes was carried out using infrared radiation (IR), nuclear magnetic resonance (NMR), electron paramagnetic resonance (EPR), and single crystal X-ray diffraction (SCXRD) and the antioxidant and cytotoxic activity of the obtained complexes was evaluated.

## 1. Introduction

Curcumin (1,7-bis-(4-hydroxy-3-methoxyphenyl)-1,6-heptadiene-3,5-diketone) is a metabolite of the Indian Curcuma species, generated by the rhizome of the perennial herb *Curcuma longa*, a member of the *gingiberacea* family widely cultivated in India and China [1]. Curcumin was isolated for the first time by Vogel and Pelletier more than two centuries ago, and Milobedzka and Lampe first proposed its chemical structure in 1910 [2,3]. Furthermore, this posed the fundamental interest that curcumin awoke gradually and expanded to other compounds that retain similar molecular topology and are denominated curcuminoids. Curcumin and curcuminoids are chelating agents due to the β–diketone functionality, which allow them to form stable complexes with a series of metal ions [1]. In recent years, numerous studies have been performed to better understand the medicinal properties of curcumin, curcuminoids and their metal complexes, with purported antitumor, antimicrobial, anti-inflammatory, antioxidant, antiviral, anti-Alzheimer and anti-cancer potential [1,4,5,6,7]. Based on the wide medicinal applications reported in the literature for copper complexes, we tried to obtain copper complexes with several curcuminoids as ligands designed to promote homoleptic structures and to investigate their biological activities. However, there are very few known crystal structures of homoleptic metal complexes of curcumin and this has been attributed to an inherent low crystallinity [1,4]. In addition, they are often insoluble in water and in most common organic solvents [1]. Such property has precluded the use of single crystal X-ray diffraction as a characterization technique [1,4,8,9,10]. As a consequence, the studies of metal complexes of curcumin and related compounds are more focused on their biological properties [8,9,10,11,12,13,14,15,16,17,18,19,20,21] rather than in detailed structural characterizations [8,9,10].

In the present work, it was possible to use several curcuminoid as ligands for complexation with copper (II) i.e., acetylated curcumin (1,7-Bis (3-methoxyl-4-acetoxy) -phenol-1,6-heptadiene-3,5-diketone, DAC) **1**, hydrogenated acetylated curcumin (1,7-Bis (3-methoxyl-4-acetoxy) -phenol-heptane-3,5-diketone, DACH_4_) **2**, methoxylated curcumin (1,7- Bis (3,4-dimethoxy) -phenol-1,6-heptadiene-3,5-diketone, DiMeOC) **3**, benzylated curcumin (1,7-Bis (3-methoxy-4-benzyl)-phenol-1,6-heptadiene-3,5-diketone) **4**, DiBncOC and bisdemethoxy-bisdehydroxy-curcumin (1,7-diphenylhepta-1,6-diene-3,5-dione, PhCurc) **5** (see Figure 1) which proved to be suitable ligands for the formation of single crystals for X-ray studies. The characterization of all synthesized homoleptic complexes was carried out using IR, NMR and EPR in liquid state, magnetic moment, MS as well as the single crystal X-ray diffraction technique. After a full characterization was carried out, their cytotoxic and antioxidant activity was evaluated.

## 2. Results and Discussion

### 2.1. IR Spectra

The IR spectrum of DAC **1** shows two bands, one of high intensity at 1755 cm^−1^ and another of very low intensity at 1795 cm^−1^ due to the free carbonyl group of the β-diketone, indicating that the compound exists mainly in the enolic form. The low-intensity band in the range 1632–1610 is attributed to the intramolecular hydrogen bridge of the enol. The band at 966 cm^−1^ that corresponds to the trans -CH=C-double bond is also observed. The IR spectra of DAC-Cu **6** show intense bands at 1514 cm^−1^ and ~484 cm^−1^ due to the interaction of metal β-diketone group from M-O vibrations. The IR spectrum of DACH_4_
**2** shows two bands at 1757 cm^−1^ (high intensity) and 1797 cm^−1^ (very low intensity) showing a small ratio of the free carbonyl group of the β-diketone, indicating that the compound exists mainly in its enolic form (see Table 1) [13].

Three bands were observed in the range of 2966-2841 cm^−1^ of low-intensity due C-H stretch. The -CH=C- band at 965 cm^−1^ was not observed. IR spectra of DACH_4_-Cu (**7**) showed strong bands due to the interaction of the metal with the β-diketone group at 1508 cm^−1^ and an additional band at ~467 cm^−1^ due to M-O vibrations. The IR spectrum of DiMeOC **3** shows the presence of two bands, one of high intensity at 1620 cm^−1^ and another with very low intensity at 1663 cm^−1^ due to the free carbonyl group of the β-diketone, revealing that the compound exists in enolic form. The trans -CH=C- band appears at 964 cm^−1^. The IR spectra of DiMeOC-Cu **8** shows strong bands due to the interaction of the metal with the β-diketone group at 1506 cm^−1^ and an additional band at ~463 cm^−1^ due to M-O vibrations. The IR spectrum of DiBncOC **4** shows the presence of two bands, one of high intensity at 1625 cm^−1^ and the other of very low intensity at 1730 cm^−1^, due to the free carbonyl group of the β-diketone. Five bands are observed in the range 849–694 cm^−1^ due C-H bending. The trans -CH=C- band appears at 970 cm^−1^. IR spectra of DiBncOC-Cu **9** shows strong bands due to the interaction of the metal with the β-diketone group at 1501 cm^−1^ and an additional band at ~465 cm^−1^ due to M-O vibrations. The IR spectrum of PhCurcu **5** shows the presence of two bands, one of high intensity at 1619 cm^−1^ and the other of very low intensity at 1670 cm^−1^, due to the free carbonyl group of the β-diketone. The trans -CH=C- band appears at 968 cm^−1^. IR spectra of PhCurcu-Cu **10** show strong bands due to the interaction of the metal with the β-diketone group at 1511 cm^−1^ and an additional band at ~420 cm^−1^ due to M-O vibrations (see Table 1) [13].

### 2.2. NMR Spectra

The ^1^H NMR spectrum of ligand DAC **1** shows one singlet for the OH proton at 16.12 ppm (strong intramolecular hydrogen bond) and one singlet for the methine proton at ~6.20 ppm (vinylic proton). Protons α to the diketone appear at 6.99 ppm and protons β to the diketone at 7.66 ppm, with a trans coupling constant of 15.9 Hz. Methoxyl and acetyl protons appear as singlets at 3.85 ppm and 2.28 ppm, respectively. The ^1^H NMR spectrum of DACH_4_ (ligand **2**) shows a keto-enol equilibrium with *ca*. 1:1 ratio. The enol tautomer shows one singlet for the OH proton at 15.53 ppm and one singlet for the methine proton at 5.78 ppm; both protons are involved in a strong intramolecular hydrogen bridge. The keto tautomer shows one singlet for the methylene proton at 3.74 ppm. Protons α to the diketone group appear at 2.77 and 2.65 ppm while protons β appear at 2.85 ppm. Methoxyl and acetyl protons appear as singlets at 3.74 ppm and 2.23 ppm, respectively. The ^1^H NMR spectrum of DiMeOC (ligand **3**) shows one singlet for the OH proton at 16.27 ppm and one singlet for the methine proton at 5.80 ppm; both protons are involved in a strong intramolecular hydrogen bond (enol tautomer). Protons α to the diketone appear at 6.47 ppm and protons β to the diketone at 7.58 ppm, with a trans coupling constant of ca. 15.8 Hz. Methoxyl protons are singlets at 3.91 ppm and 3.89 ppm. The ^1^H NMR spectrum of DiBncOC (ligand **4**) shows one singlet for the OH proton at 16.30 ppm and one singlet for the methine proton at 6.11 ppm. Unsaturated protons α to the diketone group appear at 6.84 ppm and the corresponding β protons at 7.59 ppm, with trans coupling constant of 15.78 Hz. Methoxyl protons are singlets at 3.84 ppm. Benzyl protons are singlets at 5.14 ppm. The ^1^H NMR spectrum of PhCurcu (ligand **5**) shows one singlet for the OH proton at 16.11 ppm and one singlet for the methine proton at 6.21 ppm). Protons α to the diketone function appear at 6.96 ppm and protons β at 7.67 ppm, with a trans coupling constant of 15.97 Hz. The ^1^H NMR spectra of complexes **6**–**10** lack signals for the enol proton at ca. 16 ppm, due to paramagnetic effects (see Table 2)**.**

### 2.3. EPR Spectra

The EPR spectra of ligands show diamagnetic spectra, while the EPR spectra of copper complexes of ligands **6**–**10** show a typical four lines pattern (see Figure 2). The g_‖_, g_┴_, A_‖_ and A_┴_ values were obtained directly from the EPR spectra. The g_‖_ and g_┴_ values of complexes **6**-**10** were *ca*. 2.29 and 2.06, resulting from unpaired electrons in the d_x2−y2_ molecular orbital [13]. The values of g_‖_ greater than 2.3, suggest a ionic environment for the complexes. The A_‖_ values ca. 160 × 10^−4^ cm^−1^ are consistent with a typical monomeric distorted square planar geometry. The quotient g_‖_/A_‖_ provides an index of departure from the tetrahedral structure. The quotient values that fall in the range 105–135 cm^−1^ suggest a regular square planar structure, although the observed values (141–146 cm^−1^) are indicative of a strong distortion from planarity (see Table 3) [13,22,23,24]. The magnetic moment values for Cu(II) complexes suggest that they are paramagnetic with µ_effect_ values of *ca*. 2.0 B.M with one unpaired electron (see Table 3).

### 2.4. Single Crystal X-Ray Diffraction

A rigorous analysis of the crystal structure determinations of the five complexes **6**, **7**, **8**, **9a** and **10** and the triclinic polymorph **9b** (see Figure 3) reveals a four-fold coordination around copper atom with a square planar geometry (see Appendix A), which is confirmed by the characteristics of the EPR spectra. However, if we take into consideration the “close contacts” (26.8 and 33.9% longer than the average Cu-O distances) compound, **6** and **7** can be described as 4+1 (tetragonal pyramid) and 4 + 2 (octahedron), respectively. In all compounds, the transition metal resides in a special position, except for the triclinic polymorph **9**b where the copper atom resides in a general position. The curcuminoid ligands adopt a fully extended and almost planar conformation (see Appendix A). It is noteworthy that these structural variations place the methyl substituents on phenols, face to face towards the inner part of the molecule in complexes **6** and **9**b; all pointing outside in complex **6** and two pointing towards the inner and two pointing towards the outside in complexes **8** and **9**a, revealing high conformational degrees of freedom around the (1,7-bis-(4-hydroxy-3-methoxyphenol) moiety and giving an approximately overall symmetry C_2v_ for complexes **6**, **7**, **9**b and **10**, while for complexes **8** and **9**a the C_i_ symmetry gives the best description.

### 2.5. Inhibition of Lipoperoxidation (LP) in Rat Brain Homogenate

DAC proved to be the only ligand possessing antioxidant potential comparable to curcumin. The vinyl groups are essential for the antioxidant activity of the curcumin and curcuminoids, as it can be appreciated by the significant decrease in TBARS inhibitory percentage when going from DAC to DACH_4_ (see Table 4). In this case, when ether groups replace the phenolic groups of curcumin, a significant decrease in the antioxidant potential is observed. Three copper complexes **6**-**8** showed increased antioxidant activity on the lipoperoxidation in rat brain homogenate model [25,26]. Only copper complex **9** showed a decrease in antioxidant effect respect to its ligand (see Table 4). The metal complexes **7** and **8** showed high antioxidant activity on the lipoperoxidation of rat brain homogenate model with values similar α-tocopherol (see Table 5).

LP is a process initiated and mediated by reactive oxygen species (ROS), hydroxyl (HO˙), peroxyl (ROO˙), alkoxyl (RO˙) and hydroperoxyl (HOO˙) radicals and it is known that the hydroxyl radical is an important initiator in lipid peroxidation, while peroxyl and alkoxyl radicals are intermediates in the propagation phase of lipid peroxidation [27]. The results shown in Table 5 suggest that the compounds exert moderate to good inhibition of ROS. The data show that Inhibitory Concentration-50 (IC_50_) of the free ligands DACH_4_ and DiMeOC are ca. half as large (less active) than those observed for DACH_4_-Cu and DiMeOC-Cu (see Table 5), indicating that the copper complexation leads to higher activity in reducing lipid peroxidation. In general, copper is a good inducer of oxidative stress in its free form when it has the correct oxidation state. In the complexes studied in this work, copper has an oxidation state different to +1, and in our experimental conditions (pH 7.4), copper remains chelated by the ligand. Thus, the IC_50_ values support the idea that copper remains bound to the ligands. For complexes, the IC_50_ value is *ca*. half of the ligands, indicating that copper is bound to two ligand molecules. This suggests that copper is not free and does not participate as an inducer of lipid peroxidation.

### 2.6. Cytotoxic Activity

DiMeOC is the only ligand possessing higher cytotoxicity [28] than curcumin, a noteworthy fact since the phenolic groups are blocked. A significant decrease in cytotoxic activity occurs when going from DAC to DACH_4_, indicating the importance of conjugated double bonds in the heptanoid fragment of curcumin or curcuminoids. In general, a significant decrease in the cytotoxic activity of the copper complexes **6**–**10** with respect to their free ligands is observed. Interestingly compounds **6**–**10** did not show significant cytotoxic effect against the cell lines tested. Although there are several reports of cytotoxic or antitumor activity of copper complexes with curcumin, this effect appears to be related to the presence of free phenolic groups (see Table 6) [11,17,29].

## 3. Materials and Methods 

All chemicals were available commercially, and the solvents were purified by conventional methods prior to use [30]. Curcumin was obtained from natural source by usual extractive procedures and purified by crystallization.

### 3.1. Physical Measurements

Melting points were determined on an Electrothermal Engineering IA9100X1 melting point apparatus and are uncorrected. 

### 3.2. Spectroscopic Determinations

IR absorption spectra were recorded in the range of 4000–230 cm^−1^ as KBr pellets on a BRUKER Tensor 27 spectrophotometer. ^1^H and ^13^C NMR spectra were recorded in dimethyl sulfoxide (DMSO-*d_6_*) on a Bruker Fourier 300 MHz and Varian Unity Inova 500 MHz spectrometer using TMS as internal reference. The EPR spectra were recorded in DMF at liquid nitrogen temperature (77 K) on an Electron Paramagnetic Resonance Spectrometer JEOL, JES-TE300, ITC Cryogenic System, Oxford. Magnetic moments were determined using a Johnson-Matthey magnetic susceptibility balance type msb model mk II 13094-3002, with the Gouy method at room temperature. Mass spectra were recorded in a JEOL, SX 102 A spectrometer on Bruker Microflex equipped with MALDI-Flight time. Single-crystal X-ray diffractions (SCXRD) were obtained in a Bruker diffractometer, model Smart Apex, equipped with Mo radiation (λ = 0.71073Å), CCD two-dimensional detector and low-temperature device. Data collection and data reduction were performed by APEX and SAINT-Plus programs [31]. These structures were solved by direct methods using SHELX-2013 software and refined by Full-matrix least-squares procedure on F2 using SHELX-2008 program [32].

### 3.3. Inhibition of Lipid Peroxidation on Rat Brain

#### 3.3.1. Animals

Adult male Wistar rats (200–250 g) were provided by Instituto de Fisiología Celular, Universidad Nacional Autónoma de México (UNAM). Procedures and care of animals were conducted in conformity with the Mexican Official Norm for Animal Care and Handling NOM-062-ZOO-1999. They were maintained at 23 ± 2 °C under a 12/12 h light-dark cycle with *ad libitum* access to food and water.

#### 3.3.2. Rat Brain Homogenate Preparation

Animal sacrifice was carried out avoiding unnecessary pain. Rats were sacrificed with CO_2_. The cerebral tissue (whole brain), was rapidly dissected and homogenized in phosphate-buffered saline (PBS) solution (0.2 g of KCl, 0.2 g of KH_2_PO_4_, 8 g of NaCl, and 2.16 g of NaHPO_4_._7_H_2_O/L, pH adjusted to 7.4) as described elsewhere to produce a 1/10 (*w*/*v*) homogenate [33,34]. The homogenate was centrifuged at 800 rcf (relative centrifugal field) for 10 min. The supernatant protein content was measured using Folin and Ciocalteu’s phenol reagent [35] and adjusted with PBS at 2.666 mg of protein/mL.

#### 3.3.3. Induction of Lipid Peroxidation and Thiobarbituric Acid Reactive Substances (TBARS) Quantification

As an index of lipid peroxidation, TBARS levels were measured using rat brain homogenates according to the method described by Ng and co-workers [36], with some modifications. Supernatant (375 µL) was added with 50 µL of 20 µM EDTA and 25 µL of each sample concentration dissolved in DMSO (25 µL of DMSO for the control group) and incubated at 37 °C for 30 min. Lipid peroxidation was started adding 50 µL of freshly prepared 100 µM FeSO_4_ solution (final concentration 10 µM) and incubated at 37 °C for 1h. TBARS measurements were obtained as described by Ohkawa and co-workers [37], with some modifications. 500 µL of TBA reagent (0.5% 2-thiobarbituric acid in 0.05 N NaOH and 30% trichloroacetic acid, in 1:1 ratio) was added to each tube and the final suspension cooled on ice for 10 min, centrifuged at 13,400 rcf for 5 min and heated at 80 °C in a water bath for 30 min. After cooling at room temperature, the absorbance of 200 µL of supernatant was measured at λ = 540 nm in a Microplate Reader Synergy/HT BIOTEK Instrument, Inc., Winooski, VT, USA. The concentration of TBARS was calculated by interpolation on a standard curve of tetra-methoxypropane (TMP) as a precursor of MDA [37]. Results are expressed as n moles of TBARS per mg of protein. The inhibition ratio (IR [%]) was calculated using the formula IR = (C–E) × 100/C, where C is the control absorbance, and E is the sample absorbance. Butylated hydroxytoluene (BHT) and α–tocopherol were used as positive standards. All data are presented as mean ± standard error (SEM). Data were analyzed by one-way analysis of variance (ANOVA) followed by Dunnett’s test for comparison against control. Values of *p* ≤ 0.05 (*) and *p* ≤ 0.01 (**) were considered statistically significant.

### 3.4. Citotoxic activity in Human Tumor Cells

Citotoxicity of all compounds was tested against six cancer cell lines: U251 (human glioblastoma cell line), PC-3 (human caucasian prostate adenocarcinoma), K562 (human caucasian chronic myelogenous leukaemia), HCT-15 (human colon adenocarcinoma), MCF-7 (human mammary adenocarcinoma) and SKLU-1 (human lung adenocarcinoma). Cell lines were supplied U.S. National Cancer Institute (NCI). The cell lines were cultured in RPMI-1640 medium supplemented with 10% fetal bovine serum, 2 mL -glutamine, 10,000 units/mL penicillin G sodium, 10,000 µg/mL streptomycin sulfate, 25 µg/mL amphotericin B (Invitrogen/Gibco™, Thermo Fisher Scientific, Waltham, MA, USA), and 1% non-essential amino acids (Gibco). They were maintained at 37 °C in a humidified atmosphere with 5% CO_2_. The viability of the cells used in the experiments exceeded 95% as determined with trypan blue. The human tumor cytotoxicity was determined using the protein-binding dye sulforhodamine B (SRB) in microculture assay to measure cell growth, as described in the protocols established by the NCI [38,39,40]. 

### 3.5. Synthesis of Compounds

General synthetic procedure for DAC **1**, DACH_4_
**2**, DiMeOC **3**, DiBncOC **4**, DAC-Cu **6**, DACH_4_-Cu **7**, DiMeOC-Cu **8** and DiBncOC-Cu **9** is shown in Scheme 1.

Compound **1.** 4 g of curcumin in 70 mL of dichloromethane (CH_2_Cl_2_) was reacted with 2.6 mL of pyridine (Py) and 1.6 mL of acetic anhydride (Ac_2_O) at room temperature for approximately 3 h. Follow up was done by TLC. Removal of solvent from the reaction was done under reduced pressure, and the product was extracted three times with ethyl acetate (AcOEt)-water (H_2_O) in a 3:7 proportion of until pyridine was eliminated from the organic phase. The product recrystallized in AcOEt (**1**) with 70.1% yield. ^1^H NMR (600 MHz DMSO-*d_6_*): δ 2.28 (s, 6H), 3.85 (s, 6H), 6.20 (s, 1H), 6.99 (d, 2H_vinyl_, J 15.9 Hz), 7.16 (d, 2H_aryl_, J 8.1 Hz), 7.33 (dd, 2H_aryl_, J 8.2; 1.9 Hz), 7.52 (d, 2H_aryl_, J 2 Hz), 7.66 (d, 2H_vinyl_, J 15.9 Hz), 16.1 (br s, 1H,) ppm, ^13^C NMR (^13^C {^1^H} 150 MHz, DMSO-*d_6_*): δ 20.17 (C-H), 55.72 (C-H), 101.62 (C-H), 111.93 (C_aryl_), 121.27 (C_aryl_), 123.23 (C_aryl_), 124.58 (C_vinyl_), 133.59 (C_aryl_), 139.77 (C_vinyl_), 140.93 (C_aryl_), 151.11(C_aryl_), 168.30 (C=O), 183.10 (C=O) ppm, IR 1755 cm^−1^, 1596 cm^−1^, 1506 cm^−1^, 1295 cm^−1^, 1154 cm^−1^, 619 cm^−1^, MS: M+ 453.15; yellow crystals, m.p. 170.5 °C.

Compound **2.** 3.8 g of DAC dissolved in 60 mL of AcOEt was reacted in hydrogen atmosphere with 380 mg of Pd/C-10%. The reaction mixture was stirred at room temperature until disappearance of the starting material was complete followed by TLC. The completion of reaction was achieved after 4 h and the reaction was filtered-off through celite; the solvent removed in vacuo. The product was purified by SiO_2_ column chromatography eluting with a 7:3 hexane-AcOEt solvent mixture and the product was dried under high vacuum (2), 80.4% yield. ^1^H NMR (500 MHz DMSO-*d_6_*): δ 2.23 (s, 12H), 2.65 (t, 4H_aliph_, J 7.86 Hz), 2.78 (m, 4H_aliph_), 2.85 (t, 8H_aliph_, J 7.02 Hz), 3.92 (d, 14H), 5.78 (s, 1H), 6.76 (dd, 2H_aryl_, J 8.09; 1.83 Hz), 6.79 (dd, 2H_aryl_, J 8.09; 1.83 Hz), 6.93 (s, 1H_aryl_), 6.95 (d, 2H_aryl_, J 2.44 Hz), 6.96 (m, 3H_vinyl_, Hz), 7.00 (d, 2H_aryl_, J 1.83 Hz), 15.51 (br s, 1H,) ppm. ^13^C NMR (^13^C {^1^H} 125 MHz, DMSO- *d_6_*): δ 20.17 (C-H), 29.03 (C_aliph_), 31.02 (C_aliph_), 39.44 (C_aliph_), 39.50 (C_aliph_), 44.72 (C_aliph_), 56.19 (C-H), 56.65 (C_aliph_), 100.11 (C-H), 113.25 (C_aryl_), 113.32 (C_ary_l), 120.49 (C_aryl_), 120.54 (C_aryl_) 122.90 (C_aryl_), 122.94 (C_aryl_), 137.91 (C_vinyl_), 138.03 (C_aryl_), 140.05 (C_vinyl_), 140.35 (C_aryl_), 151.02 (C_aryl_), 169.07 (C=O), 193.57 (C=O), 204.87 (C=O) ppm, IR 2974 cm^−1^, 2939 cm^−1^, 2841 cm^−1^, 1795 cm^−1^, 1757 cm^−1^, 1597 cm^−1^, 1510 cm^−1^, 1271 cm^−1^, 1188 cm^−1^, 599 cm^−1^, 526 cm^−1^, 469 cm^−1^, MS: M+ 455.85; white solid, m.p. 68.3 °C.

Compound **3.** 4 g of curcumin in 120 mL of anhydrous acetone was reacted with 0.75 g of potassium carbonate (K_2_CO_3_) and 1.2 mL of dimethyl sulfate (SO_2_(OCH_3_)_2_) and refluxed with stirring for 48 h until disappearance of the starting material. The solvent was removed under reduced pressure and the product was extracted with a 3:7 mixture of ethyl acetate AcOEt-water (H_2_O) and NaOH 10% until SO_2_(OCH_3_)_2_ was removed from the organic phase. The product was purified by SiO_2_ column chromatography eluting with a 5:4.5:0.5 mixture of hexane-CH_2_Cl_2_-MeOH and the product was dried under high vacuum (3), 63.2% yield. The product was recrystallized in MeOH, 65.4% yield. ^1^H NMR (500 MHz DMSO-*d_6_*): δ 3.89 (s, 6H,) 3.91 (s, 6H), 5.80 (s, 1H), 6.47 (d, 2H_vinyl_, J 15.76 Hz), 6.85 (d, 2H_aryl_, J 8.34 Hz), 7.11 (dd, 2H_aryl_, J 8.33; 2.00 Hz), 7.05 (d, 2H_aryl_, J 2.02 Hz), 7.58 (d, 2H_vinyl_, J 15.80 Hz), 16.27 (br s, 1H,) ppm. ^13^C NMR (^13^C {^1^H} 125 MHz, DMSO-*d_6_*): δ 60.74 (C-H), 106.16 (C-H), 115.64 (C_aryl_), 116.84 (C_aryl_), 127.20 (C_aryl_), 128.07 (C_aryl_), 132.73 (C_vinyl_), 145.57 (C_aryl_), 154.19 (C_aryl_), 156.14 (C_aryl_), 188.37 (C=O) ppm, IR 3005 cm^−1^, 2926 cm^−1^, 2831 cm^−1^, 1624 cm^−1^, 1583 cm^−1^, 1504 cm^−1^, 1134 cm^−1^, 802 cm^−1^, 607 cm^−1^, 559 cm^−1^, 544 cm^−1^, 469 cm^−1^, MS: M+ 396.77; orange crystals, m.p. 133.5 °C.

Compound **4.** 4 g of curcumin in 120 mL of anhydrous acetone was reacted with 0.75 g of K_2_CO_3_ and 2.6 mL of benzyl bromide (BnBr) at reflux for approximately 30 h until the disappearance of the starting material by TLC. The reaction solvent was removed under reduce pressure. The product was purified by SiO_2_ column chromatography eluting with a 7:3 hexane-AcOEt solvent mixture. The product was crystallized in AcOEt, 58.7% yield. ^1^H NMR (500 MHz DMSO-*d_6_*): δ 3.84 (s, 6H,) 5.14 (s, 4H), 6.11 (s, H), 6.84 (d, 2H_vinyl_, *J* 15.79 Hz), 7.09 (d, 2H_aryl_, *J* 8.47 Hz), 7.25 (dd, 2H_vinyl_, *J* 8.37; 2.00 Hz), 7.34 (t, 2H_aryl_, *J* 7.22 Hz), 7.38 (d, 2H_aryl_, *J* 1.94 Hz), 7.40 (t, 4H_aryl_, *J* 7.36 Hz), 7.45 (dd, 4H_aryl_, *J* 7.49; 2.19 Hz), 7.59 (d, 2H_aryl_, *J* 15.76 Hz), 16.30 (br s, 1H). ^13^C NMR (^13^C {^1^H} 125 MHz, DMSO-*d_6_*): δ 55.65 (C-H), 69.83 (C-H), 101.06 (C-H), 110.80 (C_aryl_), 113.24 (C_aryl_), 122.18 (C_aryl_), 122.70 (C_aryl_), 127.81 (C_vinyl_), 128.40 (C_aryl_), 136.71 (C_vinyl_), 140.33 (C_aryl_), 149.30 (C_aryl_), 149.91 (C_aryl_), 183.17 (C=O) ppm, IR 3061 cm^−1^, 2922 cm^−1^, 2856 cm^−1^, 1726 cm^−1^, 1628 cm^−1^, 1585 cm^−1^, 1510 cm^−1^, 1126 cm^−1^, 970 cm^−1^, 739 cm^−1^, 696 cm^−1^, 486 cm^−1^, 459 cm^−1^, MS: M^+^ 548.96; yellow crystals, m.p. 159.1 °C.

Compound **5.** PhCurcu was prepared in accordance with a previously reported synthetic method [41], 56.7% yield. ^1^H NMR (500 MHz DMSO-*d_6_*): δ 6.21 (s, 1H), 6.96 (d, 2H_vinyl_, *J* 15.97 Hz), 7.45 (m, 6H), 7.67 (d, 2H_vinyl_, *J* 15.92 Hz), 7.73 (dd, 4H_aryl_, *J* 7.67; 1.69 Hz), 16.11 (br s, 1H) ppm, yellow crystals, m.p. 140.5 °C.

Compound **6.** 1 mmol of DAC was dissolved in 30 mL of a 7:3 mixture of ethyl acetate-methanol. Then, a solution of copper acetate in MeOH and H_2_O (0.5 mmol) was added dropwise. After 2 h of stirring at room temperature, a brown powder was formed, which was filtered and crystallized with DMSO, 86.9% yield. ^1^H NMR (500 MHz DMSO-*d_6_*): δ 2.27 (s, 6H), 3.85 (s, 6H), 6.26 (br s, 1H), 6.81 (br s, 4H), 7.16 (br s, 1H), 7.34 (br s, 2H), 7.52 (br s, 1H), 7.66 (br s, 1H) ppm, IR 2975 cm^−1^, 2941 cm^−1^, 1752 cm^−1^, 1592 cm^−1^, 1514 cm^−1^, 1412 cm^−^, 1299 cm^−1^, 1156 cm^−1^, 604 cm^−1^, 484 cm^−1^, brown powder, m.p. 242.5 °C.

Compound **7.** 1 mmol of DACH_4_ was dissolved in 25 mL of a 7:3 mixture of ethyl acetate-methanol and 0.5 mmol of a MeOH/H_2_O solution of copper acetate was added slowly. After stirring at room temperature for 2 h, a blue powder was formed, which was filtered and crystallized in DMF/CH_3_CN, 86% yield. ^1^H NMR (500 MHz DMSO-*d_6_*): δ 4.01 (br s, 6H), 6.40 (br s, 2H), 7.05 (br s, 2H) ppm, IR 3020 cm^−1^, 2960 cm^−1^, 2926 cm^−1^, 2870 cm^−1^, 1759 cm^−1^, 1734 cm^−1^, 1574 cm^−1^, 1508 cm^−1^, 1196 cm^−1^, 1032 cm^−1^, 550 cm^−1^, 515 cm^−−1^, 469 cm^−1^, MS: M^+^ 973.27; blue powder, m.p. 165.8 °C.

Compound **8.** 1 mmol of DiMeOC was dissolved in 30 mL of a 7:3 mixture of ethyl acetate-methanol and 0.5 mmol of a MeOH/H_2_O solution of copper acetate was added dropwise. After stirring for 2 h at room temperature, a dark brown powder was formed, which was filtered and crystallized in DMSO, 93.2% yield. ^1^H NMR (500 MHz DMSO-*d_6_*): δ 4.01 (br s, 6), 6.93 (br s, 5H) ppm, IR 2995 cm^−1^, 2966 cm^−1^, 2931 cm^−1^, 2843 cm^−1^, 1630 cm^−1^, 1580 cm^−1^, 1500 cm^−1^, 1421 cm^−1^, 1132 cm^−1^, 1014 cm^−1^, 968 cm^−1^, 469 cm^−1^, MS: M^+^ 856.258; dark brown powder, m.p. 258.4 °C.

Compound **9.** 1 mmol of DiBncOC was dissolved in a mixture of 50 mL tetrahydrofuran (THF), later a solution of copper acetate in MeOH and H_2_O (0.5 mmol) was added slowly. After stirring at room temperature for 2 h, a brown powder was formed, which was filtered off and recrystallized from DMF, 85.7% yield. ^1^H NMR (500 MHz DMSO-*d_6_*): δ 3.86 (br s, 6H), 5.01 (br s, 5H), 6.84 (br s, 6H), 7.35 (br s, 13H) ppm, IR 3030 cm^−1^, 3001 cm^−1^, 2936 cm^−1^, 1622 cm^−1^, 1502 cm^−1^, 1132 cm^−1^, 696 cm^−1^, 498 cm^−1^, 469 cm^−1^, MS: M^+^ 1159.476; brown powder, m.p. 182.5 °C.

Compound **10.** 1 mmol of PhCurcu was dissolved in 30 mL of a 7:3 ethyl acetate-methanol mixture and 0.5 mmol of copper acetate in a MeO/H_2_O solution was added slowly. After stirring for 2 h of at room temperature, a brown powder was formed and was filtered and crystallized in DMF, 96.3% yield. ^1^H NMR (500 MHz DMSO-*d_6_*): δ 6.91 (d, 1H_vinyl_, *J* 15.97 Hz), 7.14 (br s, 4H), 7.45 (d, 4H_aryl_, *J* 5.93 Hz), 7.66 (d, 7H_vinyl_, *J* 19.32 Hz), IR 3023 cm^−1^, 1673 cm^−1^, 1620 cm^−1^, 1572 cm^−1^, 1070 cm^−1^, 641 cm^−1^, 590 cm^−1^, 510 cm^−1^, 420 cm^−1^, MS: M^+^ 698.99; brown powder, m.p. 275.3 °C.

## 4. Conclusions

The synthesis of 5 (compound **9** has two polymorphs) new homoleptic copper complexes was achieved with 5 different curcuminoid ligands, and their crystal structures reveal a four-fold coordination with square planar geometry. The copper ion did not increase the cytotoxic properties of the complexes with respect to free ligands but instead, high antioxidant activity for compounds DAC-Cu, DACH_4_-Cu and DiMeOC-Cu was found. In our results, the presence of free phenolic groups in curcumin derivatives might not be taken as the sole criterion for antioxidant or cytotoxic activity. From the pharmacological point of view, dealing with molecular species only composed by the curcuminoid and the metal atom, thus avoiding the presence of a molecular stabilizer or “spectator”, might be considered advantageous. Moreover, the synthesis of homoleptic copper complexes of curcuminoids achieved in the present work demonstrates a feasible approach for the preparation of new homoleptic complexes of curcuminoids comprising different ligands and metals.

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
