# Peer review of "A New Family of Homoleptic Copper Complexes of Curcuminoids: Synthesis, Characterization and Biological Properties"

_molecules, 2019, doi:10.3390/molecules24050910_

Reviewer 1 Report

The manuscript reports on isolation and structural charaterisation of few Cu(II) complexes with curcumin-derived ligands. The overall quallity of the manuscript is good; however  a number of issues has to be addressed prior the publication.

The novelty of work should be addressed in greater detail. There is a large number of reports on curcumin-derived complexes and the relevant literature is highly abundant. Therefore the authors should put more emphasis on explanation of novelty of their work.

The acquisition of conventional NMR spectra for Cu(II) complexes does not give meaningful results due to the paramagnetic character of the central metal ion. Hence the broadening of the NMR signals, which generally makes the interpretation of such spectra inconclusive. The spectra of Cu(II) complexes do not carry meaningful information and their removal from the manuscript and supplementary file is recommended.

Figure captions and tables contents should be edited for their english language. Some captions are in spanish rather than english language.

The EPR data are nicely tabulated (Table 1) while the remaining data ( such as IR and NMR) are only listed in the experimental details chapter. The addition of relevant tables would be welcome and would improve the clarity of the data presentation

Based on comments above I recommend the publication of this manuscript after minor corrections.

Author Response

Reviewer 1

Open Review

English language and style

( ) Extensive editing of English language and style required 
( ) Moderate English changes required 
(x) English language and style are fine/minor spell check required 
( ) I don't feel qualified to judge about the English language and style

Yes

Can be improved

Must be improved

Not applicable

Does the introduction provide sufficient   background and include all relevant references?

( )

(x)

( )

( )

Is the research design appropriate?

(x)

( )

( )

( )

Are the methods adequately described?

(x)

( )

( )

( )

Are the results clearly presented?

( )

(x)

( )

( )

Are the conclusions supported by the results?

(x)

( )

( )

( )

The manuscript reports on isolation and structural charaterisation of few Cu(II) complexes with curcumin-derived ligands. The overall quallity of the manuscript is good; however  a number of issues has to be addressed prior the publication.

The novelty of work should be addressed in greater detail. There is a large number of reports on curcumin-derived complexes and the relevant literature is highly abundant. Therefore the authors should put more emphasis on explanation of novelty of their work.

Two new lines of text have been added in abstract and conclusions to emphasize the novelty of the work.

The acquisition of conventional NMR spectra for Cu(II) complexes does not give meaningful results due to the paramagnetic character of the central metal ion. Hence the broadening of the NMR signals, which generally makes the interpretation of such spectra inconclusive. The spectra of Cu(II) complexes do not carry meaningful information and their removal from the manuscript and supplementary file is recommended.

Since these spectra goes in supplementary material and do not consume Journal space,  we suggest that it is kept for readers that might be interested. 

Figure captions and tables contents should be edited for their english language. Some captions are in spanish rather than english language.

Done

The EPR data are nicely tabulated (Table 1) while the remaining data ( such as IR and NMR) are only listed in the experimental details chapter. The addition of relevant tables would be welcome and would improve the clarity of the data presentation

Done

Based on comments above I recommend the publication of this manuscript after minor corrections.

We appreciate your valuable comments.

Reviewer 2 Report

This manuscript describes a straightforward study on the synthesis and characterization of several copper(II) complexes of curcumin-based ligands. These complexes are then subjected to biological studies. Overall, this study is carried out in a competent manner. However, there are a few revisions that will be necessary before this manuscript should be recommended for publication. These revisions are listed below:

1) In the Introduction, the authors describe the types of curcuminoid ligands that they are studying (compounds 1 – 5).  A figure showing the structures of these ligand should also be added, so that they are more clear to the reader.

2) The discussion of the EPR spectra (Section 2.3) should be expanded. For example, at what temperature and in what solvent (or solid-state?) were these spectra acquired at? How were the g and A values obtained? Were the data simulated or we these values extracted by inspection? The authors state that the g value indicates that unpaired electron is in the dx2-y2. Is there are citation for this statement? Also, what is meant by “an ionic environment in the complex”?

3) In Section 2.4, the authors refer to the copper sitting on a “specific position.” The correct term is “special position”

4) In Section 2.4, the authors also state the complexes have C2v and Ci symmetry. This statement does not seem correct. What is the basis for assigning these point groups? If anything, it appears that the complexes are closer to D2h symmetry.

5) In Figure 2, the structure of Figure 6 should omit the disorder within the coordinated DMSO ligand for clarity.

6) In Section 2.5, it seems counterintuitive that the copper complexes reduce lipid peroxidation more effectively than the free ligands. Copper, as a redox-active metal, typically increase ROS stress in cells and tissues. What is the hypothesis about this result?

7) Table 3 has too many significant figures for the errors associated with these numbers.

8) In Section 2.6 and Table 4, the concentration of compound that was used for the cytotoxicity studies need to be specified.

9) What source of curcumin did the authors use for their syntheses? Often curcumin is acquired as a mixture of species that needs to be separated.

10) Line 327 and onward. Note the typo. 1H RMN instead of 1H NMR

Author Response

Reviewer 2

pen Review

English language and style

( ) Extensive editing of English language and style required 
(x) Moderate English changes required 
( ) English language and style are fine/minor spell check required 
( ) I don't feel qualified to judge about the English language and style

Yes

Can be improved

Must be improved

Not applicable

Does the introduction provide sufficient   background and include all relevant references?

(x)

( )

( )

( )

Is the research design appropriate?

(x)

( )

( )

( )

Are the methods adequately described?

( )

(x)

( )

( )

Are the results clearly presented?

( )

(x)

( )

( )

Are the conclusions supported by the results?

(x)

( )

( )

( )

This manuscript describes a straightforward study on the synthesis and characterization of several copper(II) complexes of curcumin-based ligands. These complexes are then subjected to biological studies. Overall, this study is carried out in a competent manner. However, there are a few revisions that will be necessary before this manuscript should be recommended for publication. These revisions are listed below:

1)    In the Introduction, the authors describe the types of curcuminoid ligands that they are studying (compounds 1 – 5).  A figure showing the structures of these ligand should also be added, so that they are more clear to the reader.

Ligands graphic has been included

2)    The discussion of the EPR spectra (Section 2.3) should be expanded. For example, at what temperature and in what solvent (or solid-state?) were these spectra acquired at? How were the g and A values obtained? Were the data simulated or we these values extracted by inspection? The authors state that the g value indicates that unpaired electron is in the dx2-y2. Is there are citation for this statement? Also, what is meant by “an ionic environment in the complex”?

These data can be found in experimental section. The values for g and A were calculated directly from spectra. The reference requested has been included (now it is reference 13, where the term “ionic environment” is contained).

3)    In Section 2.4, the authors refer to the copper sitting on a “specific position.” The correct term is “special position”

Correction done

4)    In Section 2.4, the authors also state the complexes have C2v and Ci symmetry. This statement does not seem correct. What is the basis for assigning these point groups? If anything, it appears that the complexes are closer to D2h symmetry.

Explanation made as follows:

The basis for this statement resides on the bis“endo” or “endo,exo”  orientation of the methoxyl groups of both curcuminoid ligands broke the D2h symmetry which only could be applied to complex 10 if deviations of planarity are ignored. The paragraph has been re-phrased as:

“The curcuminoid ligands adopt a fully extended and almost planar conformation (see supplementary material, table S2) It is noteworthy that these structural variations place the methyl substituents on phenols, face to face towards the inner part of the molecule in complexes 6 and 9b; all pointing outside in complex 6 and two pointing towards the inner and two pointing towards the outside in complexes 8 and 9b, revealing high conformational degrees of freedom around the (1,7-bis-(4-hydroxy-3-methoxyphenol) moiety and giving an approximately overall symmetry C2v for complexes 6, 7, 9b and 10, while for complexes 8 and 9a the Ci symmetry gives the best description.”

5)    In Figure 2, the structure of Figure 6 should omit the disorder within the coordinated DMSO ligand for clarity.

Done

6)    In Section 2.5, it seems counterintuitive that the copper complexes reduce lipid peroxidation more effectively than the free ligands. Copper, as a redox-active metal, typically increase ROS stress in cells and tissues. What is the hypothesis about this result?

We agree. We have added the following paragraph:

In general, copper is a good inducer of oxidative stress in its free form when it has the correct oxidation state. In the complexes studied in this work, copper has an oxidation state different to +1, and in our experimental conditions (pH 7.4) copper remains chelated by the ligand. Thus, the IC50 values support the idea that copper remains bound to the ligands. For complexes, the IC50 value is ca. half of the ligands, indicating that copper is bound to two ligand molecules. This suggests that copper is not free and does not participate as an inducer of lipid peroxidation.

Table 3 has too many significant figures for the errors associated with these numbers.

Significant figures have been restricted to two.

7)    In Section 2.6 and Table 4, the concentration of compound that was used for the cytotoxicity studies need to be specified.

done

8)    What source of curcumin did the authors use for their syntheses? Often curcumin is acquired as a mixture of species that needs to be separated.

Source has been specified.

9)    Line 327 and onward. Note the typo. 1H RMN instead of 1H NMR

Correction made.

We appreciate the suggestion/correctin made.

Reviewer 3 Report

Thank you for your paper.

Here are my remarks:

    Each product has a number noted in bold.

    Sometimes it is written in bold as in line 50 or 115

    Sometimes it is written in bold and between parentheses as in line 61 or 89.

    Use only one way to rate the product in the whole article.

Ligne 134 

    "triclinic polymorph 9a "   to write triclinic polymorph 9b

Line 142

    "for complexes 8 and 9b"  to write for complexes 8 and 9a

Line145

    "in complexes 8 and 9b"  to write in complexes 8 and 9a

Line 188

    Table 2  "Muestra"  to write Samples or Products

Line 291

    "Scheme1"  to write Scheme 1

Line 303

    "m.p. 170.50oC"  to write m.p. 170.5 °C

Line 318

    "m.p.: 68.3°C"  to write m.p. 68.3 °C

Line 332

    "m.p. 133.5°C"  to write m.p. 133.5 °C

Line 345

    "m.p.159.1°C"  to write m.p. 159.1 °C

Line 363

    "Br s"  to write br s  (Three times)

Line 369

    idem (two times)

Line 376

    idem (three Times)

Line 377

    idem (one time)

Best regards.

Author Response

Reviewer 3

Open Review

English language and style

( ) Extensive editing of English language and style required 
( ) Moderate English changes required 
( ) English language and style are fine/minor spell check required 
(x) I don't feel qualified to judge about the English language and style

Yes

Can be improved

Must be improved

Not applicable

Does the introduction provide sufficient   background and include all relevant references?

(x)

( )

( )

( )

Is the research design appropriate?

(x)

( )

( )

( )

Are the methods adequately described?

(x)

( )

( )

( )

Are the results clearly presented?

(x)

( )

( )

( )

Are the conclusions supported by the results?

(x)

( )

( )

( )

Thank you for your paper.

Here are my remarks:

Each product has a number noted in bold.

    Sometimes it is written in bold as in line 50 or 115 (corrected)

    Sometimes it is written in bold and between parentheses as in line 61 or 89.

(corrected)

    Use only one way to rate the product in the whole article.

Ligne 134 

    "triclinic polymorph 9a "   to write triclinic polymorph 9b (corrected)

Line 142

    "for complexes 8 and 9b"  to write for complexes 8 and 9a (corrected)

Line145

    "in complexes 8 and 9b"  to write in complexes 8 and (corrected)

Line 188

    Table 2  "Muestra"  to write Samples or Products (corrected)

Line 291

    "Scheme1"  to write Scheme 1 (corrected)

Line 303

    "m.p. 170.50oC"  to write m.p. 170.5 °C (corrected)

Line 318

    "m.p.: 68.3°C"  to write m.p. 68.3 °C (corrected)

Line 332

    "m.p. 133.5°C"  to write m.p. 133.5 °C (corrected)

Line 345

    "m.p.159.1°C"  to write m.p. 159.1 °C (corrected)

Line 363

    "Br s"  to write br s  (Three times) (corrected)

Line 369

    idem (two times) (corrected)

Line 376

    idem (three Times) (corrected)

Line 377

    idem (one time) (corrected)

Best regards.

Thank you for your valuable suggestions.